# The social network of medical case managers, housing providers, and health department staff in the Ryan White HIV/AIDS Program: A Midwest case study

**Joseph S. Lightner**[1,2]*, **Jamie R. Shank**[2], **Ryan McBain**[3], **Tyler Prochnow**[4]

**1** School of Nursing and Health Studies, University of Missouri - Kansas City, Kansas City, Missouri, United States of America, **2** HIV Services, Kansas City Health Department, Kansas City, Missouri, United States of America, **3** RAND Corporation, Boston, Massachusetts, United States of America, **4** Department of Health Promotion, Baylor University, Waco, Texas, United States of America

* lightnerj@umkc.edu

**Data Availability Statement:** All relevant data are within the paper and its supporting information files.

## Abstract

### Background

Broad networks of providers in the Ryan White system are essential to end the HIV epidemic. Yet, there is little evidence that has assessed social networks of providers within HIV care networks. The purpose of this study is to provide a cross-sectional analysis of the role medical case managers (MCM), housing providers (HP), and health department staff (HDS), play in a Ryan White-funded area.

### Methods

All MCMs, HPs, and HDS (N = 57) in a Midwest Transitional Grant Area were invited to participate in a social network survey. Social network composition scores, exponential random graph modeling and ANOVA tests were conducted using SPSS and R Studio.

### Results

Communication in this network was significantly more likely between SW of the same provider type (e.g. MCMs communicating with other MCMs, β = .87, p<.001). HDS reported larger overall social networks (F(2,54) = 4.59, p = <.01), larger networks with other provider types (F(2,54) = 9.23, p<.001), and higher quality of relationships with other provider types (F(2,54) = 3.90, p<.05) than MCM or HP. HDS were more likely to communicate across the system than MCMs or HPs (β = .75, p<.001).

### Discussion

HDS play an important role in communicating across provider types in care delivery for HIV. Our results indicate that health departments represent essential agencies for broad dissemination of resources and knowledge, and may bridge communication barriers for coordination of housing support and HIV care delivery.

**Funding:** This project was funded by the U.S. Department of Health and Human Services, Health Resources and Services Administration, HIV/AIDS Bureau under prime award #U1SHA29299, subaward #9920160087. The funder did not provide any role in study design, data collection and analysis, decision to publish, or preparation of the manuscript. HRSA only provided funding for authors' salaries and research materials.

**Competing interests:** The authors have declared that no competing interests exist.

# Introduction

In 2018, 37,377 individuals were newly diagnosed with HIV [1]. Most new diagnoses are of marginalized individuals who experience structural issues, such as housing, unemployment, and substance use [2]. "Ending the HIV Epidemic: A Plan for America" is a $291 million initiative by multiple federal agencies to reduce new HIV infections by 75% by 2021 and by 90% by 2030. In 2020, the U.S. federal government provided $2.39 billion to HIV treatment and prevention through the Ryan White Program [3]. Recently, Ryan White HIV/AIDS Program (RWHAP) Part F Special Projects of National Significance funded interventions to improve cross sector service coordination between healthcare and housing in an attempt to improve care and reduce new diagnoses [4]. In the HIV care continuum, case management and housing providers are commonly connected and that these connections may lead to higher rates of ART adherence and viral suppression [5].

Health departments play an integral role in care coordination for HIV treatment and prevention [6] and are highlighted as an essential member of Ending the Epidemic Campaign [7]. However, Khosla [8] noted that there seemed to be a lack of governmental agencies in the HIV care continuum network. Health departments may help fill this space by serving as hubs of community health, conducting community health assessments and supporting community health improvement plans. Health departments may be essential coordinating bodies for supportive services that are integral to preventing certain health conditions, including, HIV transmission [9]. However, to date, there has been little empirical evidence to quantify the relationships that health department staff develop and the specific role that health department fill in solving large issues, such as treatment and prevention of HIV.

Social network analysis (SNA) can be used to understand the role and impact of health department social connections and social structure [10]. SNA allows researchers to draw connections between individuals and postulate how these connections may impact their behaviors [10,11]. Typically, SNA can involve one of two approaches: personal networks (egocentric analysis) or whole networks (sociocentric analysis) [10,11]. In brief, egocentric network analysis allows researchers to examine the immediate network connections (alters) of an individual (ego) and the composition of one's personal network [11]. Egocentric analysis only represents the ego's perception of their network and may not fully represent the network at large [11]. Sociocentric network analysis involves surveying a whole network and attempting to collect all of the connections between the individuals in that network [10]. Sociocentric analysis aims to create a complete picture of the network by combining all connections and perceptions within a given bounded group. Combining both approaches may provide more insight into the communication networks of medical case managers, housing providers and local health department workers.

Several authors suggest the need to employ social network analysis in understanding connectivity between providers [12]. Most of the past studies that have used SNA were conducted at the organizational-level [8,13]. Kanamori, et al., [13] assessed relationships between health agencies who treat PLWH and social service agencies that aim to prevent new infection through distribution of condoms and HIV prevention material to conclude that such collaborations are essential to stop the HIV epidemic. While these collaborations are essential, the ties between HIV prevention, care, and treatment organizations seem to be weak [8]. Granovetter [14] suggests that weak ties may be integral to the distribution of information, knowledge, and resources. However, to date, no known studies have assessed how individuals within organizations in HIV care systems interact.

Little is known regarding the social role that health department and housing provider staff play in service coordination. The purpose of this article is to describe the social networks of

medical case managers, housing providers, and health department administrators in a mid-sized, Midwest, U.S. city. We aim to describe the unique role of health department staff within a Ryan White-funded area and understand how health departments may be essential organizations to improve service coordination.

## Methods

### Sample

Data were collected from medical case managers, housing providers, and administrators of the Ryan White system between June and July 2018. All providers in the Ryan White Transitional Grant Area (TGA) were invited to participate in an online survey via email to evaluate elements of service coordination within the system. The survey was open for approximately one month, with two reminder emails sent in weekly intervals. Participants had the option to report their email address to receive a $5 gift card. All procedures had Institutional Review Board approval by the University of Missouri, Kansas City. A total of 57 people participated in the survey. Five medical case managers did not respond to the survey. The final response rate was 91.9%.

### Measures

**Egocentric network measures.** Respondents (egos) identified up to ten people (alters) who were important to them in their professional network around housing and healthcare. For each alter they identified, they also reported where the alter worked, how often they interacted with the alter (frequency), how much they valued the alter's input (value), how long they have known the alter (duration), and how well they knew the alter (quality). Each alter was then classified as "same" or "other" based on whether they performed the same role (Housing Provider, Medical Case Manager, or Health Department Staff). Network composition scores were calculated separately for alters with the same role and alters with other roles. Network composition scores signify the total value present within the ego's network. For example, the total amount of years the ego reported knowing all alters in the network.

**Whole network measures.** Respondent nominations were also used to create a whole network. Names and initials were matched across surveys in order to create ties between individuals in the network. This network would presumably represent the professional network of communication between housing providers, medical case managers and health department staff within this Ryan White System.

### Data analysis

Normality of network composition score distribution was determined using Shapiro-Wilk tests. Several of the network composition score distributions were deemed not normally distributed as determined by a significant Shapiro-Wilk test score. Network composition scores for normally distributed variables were compared across roles using ANOVA. Kruskal-Wallis tests were used to compare network composition scores across roles where scores were not normally distributed. Total frequency, total duration, total quality, and other employment role quality were deemed normally distributed and hence were evaluated using ANOVA. For all other network composition scores medians, interquartile ranges, and Kruskal-Wallis tests were used. Mean scores, standard deviations, frequencies, medians, interquartile ranges, ANOVA, and Kruskal-Wallis tests were all done using SPSS v. 25 [15].

Exponential random graph modeling (ERGM) was used to determine significant factors associated with the presence of a connections between professionals in this network [16].

ERGMs use iterative Markov chain Monte Carlo algorithms to approximate the maximum likelihood estimates for log-odds between given parameters (factors related to network structure or characteristics of the individuals in the network) and tie presence [16]. Simply, ERGMs are simulations based on empirical data and parameters set by the researcher which help determine the probability of certain network properties occurring [16]. In this study, ERGM was fit using structural components (edges, reciprocity, and transitivity), homophily (sameness) of role, popularity (amount of connections sent or received) by role and covariates such as age, gender, education, race/ethnicity, and years in current position. Homophily is the propensity of people to associate with others that are similar to themselves (i.e., a medical case manager being connected to another medical case manager) [17]. ERGMs return parameter estimates (PE) and standard errors (SE) for each factor entered into the model. A factor is deemed significant at a $p<.05$ level if the PE is greater than two times the SE. ERGM was performed using the ergm and statnet packages in R Studio [18].

## Results

### Sample characteristics

Respondents in this study were mostly made up of medical case managers (77.2%). On average respondents were 35.9 years old (sd = 8.6) and were in their current position for 2.8 years (sd = 2.5). The majority of respondents had a graduate degree (59.6%) and were white (59.6%). Table 1 contains complete sample characteristics separated by respondent role.

### Network composition scores

There was a significant difference in the total amount of alters identified depending on the respondent's role $\chi^2(3) = 9.26$, p = .01. There were no other significant effects associated with role for total network composition characteristics as well as same role network composition characteristics. However, there were significant effects associated with role for other role network composition characteristics. There were significant differences in the total amount of alters identified of other roles $\chi^2(3) = 8.40$, p = .01, total frequency of contact with alters in other roles $\chi^2(3) = 6.12$, p = .05, and total quality of relationships with alters of other roles;

**Table 1. Sample characteristics separated by respondent role.**

|  | Housing Provider (n = 8; 14.0%) | Health Department (n = 5; 8.8%) | Medical Case Manager (n = 44; 77.2%) | Total Sample (n = 57) |
|---|---|---|---|---|
| **Gender–N(%)** | | | | |
| Male | 1 (12.5%) | 1 (20%) | 6 (13.6%) | 8 (14.0%) |
| Female | 7 (87.5%) | 4 (80%) | 37 (84.1%) | 48 (84.2%) |
| **Race/Ethnicity–N(%)** | | | | |
| Asian | 0 (0%) | 0 (0%) | 1 (2.3%) | 1 (1.8%) |
| Black or African American | 3 (37.5%) | 1 (20%) | 9 (20.5%) | 13 (22.9%) |
| Latino | 1 (12.5%) | 2 (40%) | 5 (11.3%) | 8 (14.0%) |
| White | 4 (50%) | 2 (40%) | 28 (63.6%) | 34 (59.6%) |
| **Education–N(%)** | | | | |
| College Degree | 4 (50%) | 1 (20%) | 11 (25%) | 16 (28.1%) |
| Some Graduate | 2 (25%) | 0 (0%) | 4 (9.1%) | 6 (10.5%) |
| Graduate Degree | 2 (25%) | 4 (80%) | 28 (63.6%) | 34 (59.6%) |
| **Age–M(SD)** | 39.5 (11.6) | 40.8 (17.9) | 34.6 (6.2) | 35.9 (8.6) |
| **Years in Position–M(SD)** | 1.4 (1.4) | 2.7 (2.6) | 3.1 (2.6) | 2.8 (2.5) |

**Table 2. Network composition score descriptive statistics, Kruskal-Wallis, and ANOVA results.**

| | Housing Provider | Health Department | Medical Case Manager | Test-stat | p-value |
|---|---|---|---|---|---|
| **ANOVA Tested–M(SD)** | | | | | |
| **Total Frequency** | 21.24 (10.09) | 27.60 (6.23) | 24.43 (10.88) | 1.00 | .38 |
| **Total Duration** | 16.66 (12.69) | 19.22 (15.83) | 11.57 (9.43) | 0.63 | .54 |
| **Total Quality** | 21.44 (10.46) | 25.40 (12.14) | 22.57 (10.13) | 0.30 | .75 |
| **Other Quality** | 8.40 (5.42) | 16.80 (8.87) | 13.14 (9.46) | 3.90 | .03* |
| **Kruskal-Wallis–Median (Interquartile Range)** | | | | | |
| **Total Network Size** | 10.00 (5.00) | 10.00 (4.00) | 7.00 (7.00) | 9.26 | .01* |
| **Total Value** | 36.00 (36.00) | 27.00 (27.00) | 27.00 (21.00) | 1.82 | .40 |
| **Same Network Size** | 3.00 (2.00) | 3.00 (2.00) | 3.00 (4.00) | 2.15 | .34 |
| **Same Frequency** | 10.00 (5.00) | 12.00 (6.00) | 10.00 (12.00) | 0.46 | .79 |
| **Same Value** | 12.00 (8.00) | 8.00 (9.00) | 11.00 (11.00) | 1.82 | .40 |
| **Same Duration** | 3.00 (6.00) | 3.00 (6.67) | 10.00 (11.50) | 4.06 | .13 |
| **Same Quality** | 10.00 (7.00) | 8.00 (10.00) | 11.00 (13.00) | 1.29 | .52 |
| **Other Network Size** | 5.00 (6.00) | 6.00 (5.00) | 3.00 (4.00) | 8.40 | .01* |
| **Other Frequency** | 17.00 (16.00) | 20.00 (12.00) | 7.00 (7.00) | 6.12 | .05* |
| **Other Value** | 22.00 (25.00) | 12.00 (18.00) | 11.00 (12.00) | 3.53 | .17 |
| **Other Duration** | 5.00 (9.00) | 6.50 (21.25) | 4.00 (6.33) | 2.28 | .32 |

Same and other refer to the alter's role (e.g. a medical case manager choosing another medical case manager would be considered same). Mean (Standard deviation).

*—denotes significant at p<.05

F(2,54) = 3.90, p = .03 depending on the respondent's role. Descriptive statistics of all network composition characteristics as well as ANOVA and Kruskall-Wallis results can be found in Table 2.

### Exponential random graph modeling (ERGM)

ERGM was used to determine factors which may significantly influence communication ties between individuals in this network. Table 3 contains parameters included in the ERGM as well as the related estimates and standard errors. Interpretations are given next to these statistics for clarity.

## Discussion

This study aimed to show the role of health department staff in care coordination for HIV treatment in a Midwest city. We assessed a single Ryan White funded area where the health department acts as a central hub of federal pass-through funds, as well as providing information for the Ryan White system. In this network, health department staff are essential to the network and act as featured nodes that build higher quantity and quality relationships across the network, as indicated by greater overall network size, frequency of contact, and quality of relationships of relationships with medical case managers and housing providers.

Medical case managers and housing providers seem to be relatively insular to their own group. Medical case managers were more likely to be connected to other medical case managers and housing providers were more likely to be connected to other housing providers. Additionally, those individuals who had similar friends-of-friends were more likely to be connected that those who did not have similar friends-of-friends. Bridging organizational and occupational silos has long been noted as an issue impacting care coordination [19]. These data

**Table 3. Model parameter estimates and standard errors for collaborative ties within the Ryan White system.**

| Parameter | Estimate | SE | Z-score | p-value | Interpretation |
|---|---|---|---|---|---|
| Edges | -3.77 | 0.70 | -5.37 | <.01 | Communication ties were statistically sparse in this network and occur less often than at random. |
| Reciprocity | -0.28 | 0.38 | -0.74 | .46 | Communication ties in this network were no more or less likely to be mutual than what would be expected by random. |
| Transitivity [x] | 1.32 | 0.15 | 8.60 | <.01 | Communication ties in this network were significantly more likely to occur between members who also shared a common third communication partner. |
| Role (Housing Provider reference) | | | | | |
| Medical Case Manager | 0.57 | 0.12 | 4.65 | <.01 | Medical case managers were statistically more likely to have communication ties in this network as compared to housing providers. |
| Health Department | 0.89 | 0.16 | 5.45 | <.01 | Health department staff were statistically more likely to have communication ties in this network as compared to housing providers. |
| Role Homophily | 0.87 | 0.17 | 5.09 | <.01 | Communication ties were statistically more likely to be present between members who shared the same role. |
| Model Covariates | | | | | |
| Age | -0.01 | 0.01 | -1.177 | .24 | Age of the respondent had no statistically significant effect on the odds of a tie being present. |
| Amount of Years in Position | 0.01 | 0.02 | 0.763 | .45 | The amount of years a respondent had been in their current position had no statistically significant effect on the odds of a tie being present. |
| Education (Bachelor's degree reference) | | | | | |
| Some Graduate School | 0.05 | 0.16 | 0.32 | .75 | Respondents with some graduate school education were no more or less likely to be a part of a communication tie in this network when compared to those with only Bachelor's degrees. |
| Graduate Degree | -0.09 | 0.11 | -0.83 | .41 | Respondents who had a graduate degree were no more or less likely to be a part of a communication tie in this network when compared to those with only Bachelor's degrees. |
| Gender (Male reference) | -0.09 | 0.10 | -0.91 | .37 | Males were no more or less likely to be a part of a communication tie in this network when compared to females. |
| Race/Ethnicity (White reference) | | | | | |
| Black or African American | 0.06 | 0.31 | 0.19 | .85 | Participants who identified as Black or African American were no more or less likely to be a part of a communication tie in this network when compared to White participants. |
| Latino | -0.32 | 0.32 | -0.98 | .33 | Participants who identified as Latino were no more or less likely to be a part of a communication tie in this network when compared to White participants. |
| Asian | -0.14 | 0.29 | -0.50 | .62 | Participants who identified as Asian were no more or less likely to be a part of a communication tie in this network when compared to White participants. |

[x] Transitivity was modeled using geometrically weighted edgewise shared partner distribution with a decay of 0.1

[*] Parameter estimate is greater than two times the standard error which indicates a significant effect

support the need for interventions focused on bridging institutional silos among the HIV community.

Interventions focused on health department staff may have success in bridging silos. In this study, health department staff seem to play an integral role in communicating across provider types. While health department staff were very connected to other health department staff, they also had larger total network sizes that included medical case managers and housing providers, were more likely to interact with all individuals in their network more often, and reported higher quality relationships than medical case managers or housing providers. It seems that health department staff are positioned in the network to act as communicators across organization and occupation.

Housing has been a key intervention mechanism of ending the epidemic [20,21]. The HRSA, HIV/AIDS Bureau has dedicated significant financial resources and changed organizational capacity to facilitate housing for individuals as a way to treat HIV and further prevent transmission. In this study, housing providers are less likely than health department staff to communication across the network. Interventions are needed to assist housing providers in

developing social networks across provider type to improve service coordination. Increasing the capacity of housing providers to work across professional networks may lead to overall better health outcomes from all populations experiencing housing instability.

This study fills a gap in the literature by assessing individuals within organizations, adding an additional information to the complexity of social networks. Additional strengths include the use powerful statistical techniques to quantify relationships and attempts to assess a complete network. However, this study is limited by the cross-sectional nature, relatively small sample, and unique community of Kansas City, Missouri. Additionally, post-hoc corrections were not performed due to the exploratory nature of this study that was restricted by the network size. Thus, these results may not be generalizable to other health departments in the United States.

Future studies should qualitatively assess the mechanism of relationship building among health department staff, HIV medical case managers and housing providers. Additionally, longitudinal studies assessing change over time should be conducted to understand how a closed network adapts to staff turnover, and funding and organizational changes. As Ryan White and Housing and Urban Development funding mechanisms change in the upcoming years, it will be important to assess how care coordination changes for people living with HIV.

## Supporting information

**S1 File. Node attributes: Data assocaited with nodes of the network.**
(XLSX)

**S2 File. Edge list.**
(XLSX)

**S3 File. Survey.**
(DOCX)

## Author Contributions

**Conceptualization:** Joseph S. Lightner, Jamie R. Shank, Ryan McBain.

**Data curation:** Joseph S. Lightner.

**Formal analysis:** Joseph S. Lightner, Tyler Prochnow.

**Funding acquisition:** Jamie R. Shank.

**Investigation:** Joseph S. Lightner, Ryan McBain.

**Methodology:** Joseph S. Lightner, Jamie R. Shank, Ryan McBain.

**Project administration:** Joseph S. Lightner, Jamie R. Shank.

**Resources:** Joseph S. Lightner, Ryan McBain.

**Software:** Joseph S. Lightner.

**Supervision:** Joseph S. Lightner, Ryan McBain.

**Validation:** Joseph S. Lightner.

**Visualization:** Joseph S. Lightner.

**Writing – original draft:** Joseph S. Lightner.

**Writing – review & editing:** Joseph S. Lightner, Jamie R. Shank, Ryan McBain, Tyler Prochnow.

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
