## [Decision Letter · Decision Letter 0]

16 Jul 2020

PONE-D-20-18035

The Social Network of Medical Case Managers, Housing Providers, and Health Department Staff in the Ryan White HIV/AIDS Program: A Midwest Case Study

PLOS ONE

Dear Dr. Lightner,

Thank you for submitting your manuscript to PLOS ONE. After careful consideration, we feel that it has merit but does not fully meet PLOS ONE’s publication criteria as it currently stands. Therefore, we invite you to submit a revised version of the manuscript that addresses the points raised during the review process.

We look forward to receiving your revised manuscript.

Kind regards,

Nickolas D. Zaller

Academic Editor

PLOS ONE

Journal Requirements:

2. Please address the following:

- Please include additional information regarding the survey or questionnaire used in the study and ensure that you have provided sufficient details that others could replicate the analyses. For instance, if you developed a questionnaire as part of this study and it is not under a copyright more restrictive than CC-BY, please include a copy, in both the original language and English, as Supporting Information. In addition, please describe the dvelopment and validation of this tool in further detail, for example by referring to previous works and any pre-testing that took place.

- Please refer to any post-hoc corrections to correct for multiple comparisons during your statistical analyses. If these were not performed please justify the reasons. Please refer to our statistical reporting guidelines for assistance (https://journals.plos.org/plosone/s/submission-guidelines.#loc-statistical-reporting).

3.Please ensure that you include a title page within your main document. We do appreciate that you have a title page document uploaded as a separate file, however, as per our author guidelines (http://journals.plos.org/plosone/s/submission-guidelines#loc-title-page) we do require this to be part of the manuscript file itself and not uploaded separately.

4. Thank you for including your competing interests statement; "The authors have declared that no competing interests exist. "

We note that one or more of the authors are employed by a commercial company:

RAND Corporation

Reviewers' comments:

Reviewer's Responses to Questions

**Comments to the Author**

1. Is the manuscript technically sound, and do the data support the conclusions?

Reviewer #1: Yes

Reviewer #2: Yes

2. Has the statistical analysis been performed appropriately and rigorously? 

Reviewer #1: I Don't Know

Reviewer #2: Yes

3. Have the authors made all data underlying the findings in their manuscript fully available?

Reviewer #1: No

Reviewer #2: No

4. Is the manuscript presented in an intelligible fashion and written in standard English?

Reviewer #1: Yes

Reviewer #2: Yes

5. Review Comments to the Author

Reviewer #1: This is an interesting study. I am not well-versed in social network analysis (although I am interested) so I cannot confirm the appropriateness or accuracy of the analytical techniques employed (i.e. network composition scores and/or ERGM), but if the results are accurately presented then I support acceptance of your paper. However, this recommendation comes with some required edits. Your data and results are unique to the city and/or state represented. This is implied early in the paper but needs to be addressed throughout. Thus, you must change this statement in the beginning of the discussion (pg 10, lines 157-8) to read, "This study aimed to show the role of health department staff in care coordination for HIV treatment in a Midwest city." And change the following sentence to read, "We assessed a single Ryan White funded area..." I am confident you would have different results if data were collected from my state which also has a state health dept serving as a hub for HIV programs.

Reviewer #2: This study presents the social network analysis among Ryan White case managers, housing providers, and health department staff and indicates that health department staff had a wider social network across the HIV/AIDS service system.

1. The author should provide the number of invitations sent out for each role and whether the response rates varied by roles. This information is critical to test whether selection bias existed in this study. If the author had demographic characteristics of all invited individual. It will strengthen this study by exploring associations between demographics and response to survey.

2. Please test the normality of reported network compositions. If they are not normal distributed, Kruskal–Wallis test (rank test) should be used. Median and interquartile range should be reported instead of mean and stand deviation.

3. In the ERGM analysis, race/ethnicity should be added as a model covariate.

6. PLOS authors have the option to publish the peer review history of their article (what does this mean?). If published, this will include your full peer review and any attached files.

Reviewer #1: No

Reviewer #2: No

---

## [Author Response · Author response to Decision Letter 0]

27 Jul 2020

Thank you for providing feedback on this manuscript. We have incorporate all feedback. A detailed description of each point is presented in the response to reviewers document that has been uploaded.

---

## [Decision Letter · Decision Letter 1]

18 Aug 2020

The Social Network of Medical Case Managers, Housing Providers, and Health Department Staff in the Ryan White HIV/AIDS Program: A Midwest Case Study

PONE-D-20-18035R1

Dear Dr. Lightner,

We’re pleased to inform you that your manuscript has been judged scientifically suitable for publication and will be formally accepted for publication once it meets all outstanding technical requirements.

Kind regards,

Nickolas D. Zaller

Academic Editor

PLOS ONE

Reviewers' comments:

Reviewer's Responses to Questions

**Comments to the Author**

1. If the authors have adequately addressed your comments raised in a previous round of review and you feel that this manuscript is now acceptable for publication, you may indicate that here to bypass the “Comments to the Author” section, enter your conflict of interest statement in the “Confidential to Editor” section, and submit your "Accept" recommendation.

Reviewer #1: All comments have been addressed

Reviewer #2: (No Response)

2. Is the manuscript technically sound, and do the data support the conclusions?

Reviewer #1: Yes

Reviewer #2: (No Response)

3. Has the statistical analysis been performed appropriately and rigorously? 

Reviewer #1: I Don't Know

Reviewer #2: (No Response)

4. Have the authors made all data underlying the findings in their manuscript fully available?

Reviewer #1: Yes

Reviewer #2: (No Response)

5. Is the manuscript presented in an intelligible fashion and written in standard English?

Reviewer #1: Yes

Reviewer #2: (No Response)

6. Review Comments to the Author

Reviewer #1: The revisions have satisfied my comments/suggestions. I do not have any additional comments or revisions.

Reviewer #2: Authors have addressed all my concerns about statistical analysis. They conducted normality tests and used median and Internationale range to present their data. They also add race/ethnic into their final model.

7. PLOS authors have the option to publish the peer review history of their article (what does this mean?). If published, this will include your full peer review and any attached files.

Reviewer #1: No

Reviewer #2: No

---

## [Editor Report · Acceptance letter]

19 Aug 2020

PONE-D-20-18035R1 

The Social Network of Medical Case Managers, Housing Providers, and Health Department Staff in the Ryan White HIV/AIDS Program: A Midwest Case Study 

Dear Dr. Lightner:

I'm pleased to inform you that your manuscript has been deemed suitable for publication in PLOS ONE. Congratulations! Your manuscript is now with our production department. 

Kind regards, 

on behalf of

Dr. Nickolas D. Zaller 

Academic Editor

PLOS ONE